# Direct Phenotyping and Principal Component Analysis of Type Traits Implicate Novel QTL in Bovine Mastitis through Genome-Wide Association

**DOI:** 10.3390/ani11041147

**Published:** 2021-04-17

**Authors:** Asha M. Miles, Christian J. Posbergh, Heather J. Huson

**Affiliations:** Department of Animal Science, Cornell University, Ithaca, NY 14853, USA; asha.miles@usda.gov (A.M.M.); cjp98@cornell.edu (C.J.P.)

**Keywords:** genome-wide association, mastitis, principal component analysis, udder conformation, teat conformation

## Abstract

**Simple Summary:**

It is well established that the physical conformation of a cow’s udder and teats may influence her susceptibility to mastitis, an inflammatory condition of the udder, which has 25% prevalence in the United States. Our aim was to improve the biological understanding of the genetics underlying mastitis by intensively characterizing cows for udder and teat conformation, including the novel traits of teat width and end shape, and directly associating those phenotypes with high-density genotypes for those exact same cows. We also generated a composite measure that accounts for multiple high-mastitis-risk udder and teat conformations in a single index for risk phenotypes. Using this approach, we identified novel genetic markers associated with udder and teat conformation, which may be good candidates for inclusion in national genetic evaluations for selection of mastitis-resistant cows. Mastitis is the costliest disease facing US dairy producers, and integrating genetic information regarding disease susceptibility into breeding programs may be an efficient way to mitigate economic loss, support the judicious use of antimicrobials, and improve animal welfare.

**Abstract:**

Our objectives were to robustly characterize a cohort of Holstein cows for udder and teat type traits and perform high-density genome-wide association studies for those traits within the same group of animals, thereby improving the accuracy of the phenotypic measurements and genomic association study. Additionally, we sought to identify a novel udder and teat trait composite risk index to determine loci with potential pleiotropic effects related to mastitis. This approach was aimed at improving the biological understanding of the genetic factors influencing mastitis. Cows (N = 471) were genotyped on the Illumina BovineHD777k beadchip and scored for front and rear teat length, width, end shape, and placement; fore udder attachment; udder cleft; udder depth; rear udder height; and rear udder width. We used principal component analysis to create a single composite measure describing type traits previously linked to high odds of developing mastitis within our cohort of cows. Genome-wide associations were performed, and 28 genomic regions were significantly associated (Bonferroni-corrected *p* < 0.05). Interrogation of these genomic regions revealed a number of biologically plausible genes whicht may contribute to the development of mastitis and whose functions range from regulating cell proliferation to immune system signaling, including *ZNF683*, *DHX9*, *CUX1*, *TNNT1*, and *SPRY1.* Genetic investigation of the risk composite trait implicated a novel locus and candidate genes that have potentially pleiotropic effects related to mastitis.

## 1. Introduction

Mastitis, a condition characterized by inflamed mammary tissue and the udder gland, is the costliest disease facing the US dairy producers, accounting for an estimated $2 billion in annual losses and 11% of total milk lost according to a recent market analysis [1]. Due to its well-documented impact on cow health and production, mastitis research has been prioritized since the early 1900s, and as genomic tools have evolved, so have our approaches to understanding this disease [2,3,4]. While mastitis has been historically considered a management problem, genetic correlations among milk yield, mastitis susceptibility, and udder morphology encouraged selection for udder and teat type traits as early as the 1950s [5]. In 2009, udder composite values were incorporated into official national genomic evaluation systems to account for the influence of cow conformation on health traits [6,7]. While udder and teat morphologies have been established as proxy traits for mastitis susceptibility, there is little consensus in the literature regarding the exact relationship of mastitis to udder and teat type traits or their respective heritabilities [8,9]. Most extant research into cow conformation traits has relied on pedigree information to calculate relationship matrices for estimation of heritability, genetic correlation, and variance [10,11,12]. In pursuit of genetic improvement among US dairy herds, additional focus has also been given to evaluating sire transmitting abilities, and while these studies comprehensively investigate udder morphology, teat length, and teat placement, other teat characteristics such as width and end shape are neglected [11,13].

As genotyping technologies have become increasingly cost effective, more molecular genetic studies are emerging. The broad-spread popularity of artificial insemination bulls has presented an economical opportunity to conduct large genome-wide association (GWA) studies by indirectly associating bull genotypes with the performance records of their daughters. For example, a recent study using publicly available whole genome sequences to impute medium- and high-density single-nucleotide polymorphism (SNP) data of up to over 20 million sequence variants identified quantitative trait loci (QTL) for a number of udder and teat type traits in Fleckvieh cattle [14]. A testament to this approach is the overall improvement for mastitis-related traits (i.e., milk somatic cell count (SCC), udder and teat morphology) and the increase in the average rate of mastitis resistance observed among US Holsteins in the past two decades [15]. However, a potential drawback to large-scale studies that use genotype imputation from low-density SNP arrays is that while they may maximize genome coverage, they introduce bias toward population averages and potentially limit the research ability to detect rare variants [16]. In addition, while associating bull genotypes with daughter performance has been successful in identifying potential sires that possess udder conformation genetics related to mastitis resistance, the fact that bulls neither get mastitis nor develop udders may limit our ability to understand the genetic regulation of these traits with this approach. Currently, only one study has directly associated udder traits with cow genotypes, relating beef cow teat length, teat diameter, and a composite udder support score to genomic data acquired from the low-density Bovine Illumina 50 k chip [17].

A high-density GWA study of US Holstein dairy cow udder and teat conformation with direct phenotype–genotype associations and no reliance on genotype imputation has yet to be conducted. Thus, we previously conducted a prospective cohort study in which we assembled detailed udder and teat phenotypes and associated flat teat ends and loose fore udder attachment with increased odds of both an elevated SCC and clinical mastitis, as well as a low rear udder height and increasing rear teat width with higher odds of experiencing clinical mastitis alone [18]. We posited that these risk factors may be effective criteria to include in culling protocols or inform mating strategies by selecting bulls with favorable evaluations for udder and teat morphology without sacrificing milk yield. The purpose of this current study was to use GWA approaches to identify SNP markers associated with udder and teat type traits, giving special focus to these four risk traits, for incorporation into genomic selection marker panels for mastitis-resistant cows.

## 2. Materials and Methods

This study was approved by the Cornell University Institutional Animal Care and Use Committee under authorization reference number 2014-0121 on 2/20/2015, and signed farm owner consent was obtained before the commencement of this study.

### 2.1. Phenotyping

We conducted a prospective cohort study of 523 Holstein cows on 2 commercial herds in upstate New York involving direct udder and teat phenotype determination, as previously described [18]. Udder and teat traits, including fore udder attachment, udder cleft, udder depth, rear udder height, and rear udder width, as well as front and rear teat placement, end shape, length, and width, were scored by one trained researcher. Front and rear teat length and width were measured using a translucent ruler with a scale unit of 1 cm, and front and rear teat end shape were scored as either flat, round, or pointed. The remaining traits were scored on 3 levels coded as 0, 1, or 2, adapted from the US Holstein Association’s linear descriptive traits, which can be found here: https://www.holsteinusa.com/pdf/print_material/linear_traits.pdf (accessed on 1 February 2015). In addition, cows were given a binary score for each trait according to whether they had a high-mastitis-risk phenotype (e.g., loose fore udders: high risk) versus all other conformations (intermediate and tight fore udders: low risk). These determinations were made based on our previous univariate assessment of each trait’s relationship to elevated SCC or clinical mastitis [18].

### 2.2. Principal Component Analysis

Principal component analysis (PCA) of phenotype data was performed to identify a composite measure that may describe multiple udder and teat traits using R Studio version 3.2.5 (R Core Team, 2019). Four PCAs were performed: (1) teat traits only (front and rear teat placement, end shape, length, and width), (2) udder traits only (fore udder attachment, udder cleft, udder depth, rear udder height, and rear udder width), (3) teat and udder traits together, and (4) risk traits only (rear teat end shape, rear teat width, fore udder attachment, and rear udder height, based on our prior assessment of these particular cows [18]). Bartlett’s test of sphericity was performed to confirm that these data could be reduced with PCA [19]. All principal components (PCs) with an eigenvalue of >1 were assessed for their usefulness in describing udder and teat phenotypes by examining the percentage of variance they explained and their loading values for each trait of interest. The annotated code and phenotypic data used for PCA are publicly available at https://github.com/AshaMilesPhD/U-T_GWAS (created on 15 March 2021).

### 2.3. Genotyping and Quality Control

A whole blood sample was taken from each cow via the coccygeal vein, collected in 10 mL K_2_EDTA anticoagulant vacutainers (Becton, Dickinson & Company, Franklin Lakes, NJ, USA), and stored at 4 °C or −20 °C until DNA extraction. Genomic DNA was extracted from whole blood according to the Gentra Puregene Blood Kit Protocol (Gentra Systems, Inc. Minneapolis, MN, USA) using laboratory-made buffers. Only DNA from cows with complete phenotype records (N = 471) was submitted to GeneSeek (Neogen Genomics, Lincoln, NE, USA) for SNP genotyping on the Illumina BovineHD777k beadchip (Illumina, Inc., San Diego, CA, USA). Quality control filtering was applied to all genotypes via Golden Helix SNP & Variation Suite (SVS) software v8.3.4 (Golden Helix, Bozeman, MT, USA). Genotypes were retained if the SNP and individual call rate was ≥0.9, minor allele frequency was ≥0.05, and allele number was ≤2. Identity by descent (IBD) estimates were calculated for all pairs of individuals based on available genotype data, and individuals with an IBD estimate of ≥0.9 demonstrating significant relatedness were removed from the dataset [20]. After all quality control filtering was applied, 458 cows with 581,663 SNPs remained for analysis.

### 2.4. Genome-Wide Association

Efficient mixed model linear analysis (EMMAX) models were employed in Golden Helix SVS, allowing the inclusion of the IBD matrix to correct for any population structure among this cohort of cows [21]. EMMAX computes the following:(1)y=Xβ+Zu+e,
where y is an n x 1 vector of observed phenotypes; X is an n x f matrix of fixed effects, including the mean, SNP, and other covariates; β is an f x 1 vector representing the coefficient of the fixed effects; Z is an n x t matrix relating the instances of the random effect to the phenotypes; and u is the unknown random effect of the mixed model with
(2)$$Var\left(u\right)=\sigma_{g}^{2}K,$$
where $\sigma_{g}^{2}$ is the genetic component of variance, K is the kinship matrix inferred from the genotype, and e is an n x n matrix of residual effects that cannot be explained by the variables in the model [21]. In EMMAX, these variance components are calculated for the entire association analysis because the effect for each SNP is small. Additive (risk of phenotype increases with each copy of the allele), dominant (only 1 copy of the allele is required for increased risk), and recessive (2 copies must be present for increased risk of phenotype) inheritance models [22] were considered, along with the fixed effects of farm, parity, genotyping batch, rear teat length, udder depth, and rear udder width as potential covariates, given their observed correlation with and potential confounding effects on the traits of interest.

To assess continuous variation in each trait of interest, genome-wide associations (GWAs) were run for the quantitative traits of fore udder attachment, udder cleft, udder depth, rear udder height, and rear udder width, as well as front and rear teat placement, end shape, length, and width. To compare extreme differences in morphology believed to contribute to the development of mastitis, case–control GWAs were performed for each trait, with cows classified by the high-risk phenotype determination described above. To address the possibility that the genetic variance component of the phenotype is greater for cows who have had less exposure to the environment, a primiparous-only subset of the larger cohort of cows was also evaluated (*n* = 144).

All *p*-values were adjusted for multiple testing using a Bonferroni correction and the false discovery rate (FDR). Quantile–quantile (QQ) plots of the log_10_(expected *p*-values) against the log_10_(observed *p*-values) and a genomic inflation factor pseudo-lambda were used to assess goodness-of-fit, the latter calculated as follows:(3)$$\lambda =\frac{median\;observed\;P-value}{median\;expected\;P-value}$$

The pseudo-heritabilities of each trait were calculated as follows:(4)$$ph_{j}=\frac{{\hat{\sigma }}_{g}^{2}\;}{Var\left(y\right)}$$
where $\hat{\sigma_{g}^{2}}$ is the genetic component of variance and Var(y) is given by $\hat{\sigma_{g}^{2}}+\hat{\sigma_{e}^{2}}$, the sum of the genetic (g) and error (e) components of variance [23]. A stringent multiple testing correction was applied, and only regions with the SNP passing the Bonferroni correction (adjusted *p*-value < 0.05) were interrogated for candidate genes. Any gene in linkage disequilibrium (LD) with an associated marker, or in a 500 kb upstream or downstream range of said marker in the absence of LD, was identified using the National Center for Biotechnology Information RefSeq database [24]. All genome coordinates given use the most recent ARS_UCD 1.2 bovine genome assembly (https://www.ncbi.nlm.nih.gov/assembly/GCF_002263795.1/ (accessed on 30 July 2019)).

## 3. Results

### 3.1. Principal Component Analysis

Udder traits-only PC1 accounted for 41% of the variance in the phenotype and described the overall udder size, loading toward deeper, lower, and wider rear udders, as well as looser fore udder attachment. Risk traits-only PC1 accounted for 35% of the phenotypic variance, loading in the direction of the low-risk phenotypes of thinner rear teats, tighter fore udders, and higher rear udders (Figure 1A). This suggests cows that have higher risk PC1 scores have lower mastitis risk based on the proxy traits of udder and teat conformation. No other PCs were informative or considered for GWA.

### 3.2. Genome-Wide Association

Final GWA models are summarized in Table 1. The QQ plots referenced to select the final models can be found in Appendix A. All QTL positions and biologically plausible candidate genes are detailed in Appendix A. No SNPs were significantly associated with udder PC1, front teat end shape, rear teat placement, front teat placement, or udder cleft in our total cohort of cows. In our primiparous subset, SNPs were significantly associated (FDR < 0.05) with front teat placement and udder depth only.

#### 3.2.1. Risk PC1

Risk PC1 was significantly associated with one SNP at the *Bos taurus* autosome (BTA) 15:7287030 in a dominant inheritance model with covariates of farm, parity, rear teat length, udder depth, and udder width (Bonferroni-corrected *p* < 0.05; Figure 1B).

#### 3.2.2. Fore Udder Attachment

An additive case–control mixed model with no covariates comparing loose (highest mastitis risk) versus tight fore udder attachment significantly associated one SNP at BTA 2:126359098 (Bonferroni-corrected *p* < 0.05) and the X chromosome position 121461599-121477381 (Bonferroni-corrected *p* < 0.05; Figure 2).

#### 3.2.3. Udder Depth

Significant associations were observed for udder depth in both the total cow cohort and the primiparous subpopulation. In a case–control comparison of deep (highest mastitis risk) versus high udders (lowest mastitis risk) in the total cohort (N = 458), a dominant mixed model with parity and udder width as fixed effects significantly associated one SNP at BTA 5:113268242 (Bonferroni-corrected *p* < 0.05; Figure 3A). In comparison, by evaluating primiparous cows only (*n* = 144) under the hypothesis that their genetic component of variance is greater because they have had less exposure to the environment, a linear recessive mixed model significantly associated a QTL at BTA 17:34476230-34552407 (FDR-corrected *p* < 0.05; Figure 3B).

#### 3.2.4. Rear Udder Height

In a case–control comparison of low udder heights (high mastitis risk) to all others (intermediate and high udders combined: lower mastitis risk), a recessive mixed model with covariates of udder depth, udder width, and fore udder attachment significantly associated 32 SNPs (Bonferroni-corrected *p* < 0.05) at BTAs 6, 14, 15, 18, and 22 (Figure 4).

#### 3.2.5. Udder Width

In a case–control comparison of wide (high mastitis risk) versus all other udder widths (narrow and intermediate), a dominant mixed model with no covariates significantly associated one SNP (Bonferroni-corrected *p* < 0.05) at BTA 15:75722222 with udder width (Appendix A).

#### 3.2.6. Front Teat Length and Width

A linear recessive mixed model with farm, parity, front teat width, and rear teat length and width covariates associated a QTL at BTA 10:50107685-50117321 (Bonferroni-corrected *p* < 0.05) with front teat length (Appendix A). An additional five associated SNPs passed FDR correction at BTAs 9, 10, and 12. The most appropriate model for a case–control comparison of wide (higher mastitis risk) versus narrow front teats was a recessive mixed model with farm, parity, and front teat length included as fixed effects, associating four SNPs at BTA 23 (Bonferroni-corrected *p* < 0.05; Appendix A).

#### 3.2.7. Rear Teat Length, Width, and End Shape

A case–control GWA comparing short versus long (higher mastitis risk) rear teats associated one SNP (Bonferroni corrected *p* < 0.05) at BTA 2:112245780 in an additive mixed model with no covariates (Appendix A). In an examination of rear teat width, a case–control GWA comparing narrow (lower mastitis risk) versus wide (higher mastitis risk) rear teats associated one SNP at BTA 25:38568564 in a recessive mixed model with no covariates (Bonferroni-corrected *p* < 0.05; Figure 5A). An additional three SNPs passed FDR correction at BTAs 18, 25, and 28. Considering continuous variation in the rear teat width, a linear recessive GWA with no covariates associated 23 SNPs (Bonferroni-corrected *p* < 0.05) at BTAs 10, 11, 16, 18, 19, and 25 (Figure 5B). An additional 50 SNPs spanning the genome passed FDR correction. In a case–control representation of the rear teat end shape of flat (high mastitis risk) versus pointed (low mastitis risk), a recessive mixed model with no covariates associated one SNP (Bonferroni-corrected *p* < 0.05) at BTA 26:50630351 (Figure 6).

#### 3.2.8. Front Teat Placement

In the primiparous-only subset of cows, a linear GWA for front teat placement associated a QTL at BTA 9:57652720-58079933 (Bonferroni-corrected *p* < 0.05; Appendix A).

## 4. Discussion

The purpose of this study was to use GWA approaches to identify SNP markers associated with udder and teat type traits, giving special focus to traits we previously associated with increased odds of an elevated SCC and/or clinical mastitis in this same cohort of cows [18]. These studies provide new insight into the genetic regulation of teat and udder conformation and mastitis susceptibility by using direct phenotyping and by considering these phenotypes on the basis of their relationship to mastitis risk. In recognition of potential false-positive inflation due to multiple assessments of each trait (continuous variation in the trait, extremes in morphology, and only primiparous cows with less exposure to the environment), only SNPs that passed a strict Bonferroni correction were further explored. Below follows a discussion of associated QTL and candidate genes most biologically relevant to mastitis risk-associated traits.

### 4.1. Risk PC1

Our goal in performing PCA on these phenotypes was to identify a single measure that may describe multiple udder and teat phenotypes and, by extension, mastitis risk. Risk PC1 described this risk (excepting rear teat end shape), and GWA for this new composite measure significantly associated a novel QTL at 15:7287030, which was not identified by individual assessment of those risk traits (Figure 1). Furthermore, this novel QTL has not been implicated in any prior study according to the most recent release of the Cattle QTL database (August 2020) [25]. Candidate genes identified in this region were related to both cell division (*CEP126 (Centrosomal Protein 126*)) and immune cell progenitor differentiation (*ANGPTL5 (Angiopoietin like 5*), supporting our theory that risk PC1 does indeed reflect both mastitis and udder and teat morphology [26,27].

### 4.2. Fore Udder Attachment

In this same cohort of cows, we previously associated loose fore udder attachment with high odds of both an elevated SCC and clinical mastitis, suggesting that fore udder attachment is a relevant criterion on which to base culling and management decisions for mastitis control [18]. A case–control GWA for extremes in this phenotype (loose attachment, representing most at risk; tight attachment, representing least at risk) implicated a number of genes related to both immune function and cell proliferation near associated markers at 2:126354670-126359098 (Figure 2). This QTL has been previously mapped for multiple types of milk acid content, milk fat yield, as well as conception rate [28,29,30]. Among candidate genes in this region is the immunomodulator *Wiskott–Aldrich Syndrome Protein Family 2* (*WASF2*), a cytoplasmic protein implicated in cell migration, phagocytosis, and immune synapse formation [31]. Other plausible candidate genes underlying fore udder morphology include *Nuclear Distribution C* (*NUDC*), which is critical to cytokinesis; *Stratifin* (*SFN*), which may regulate cell cycle progression; and *Keratinocyte Differentiation Factor 1* (*KDF1*), which serves as an essential regulator of epidermis formation [32,33,34]. In addition, this same genomic region houses genes related to immune function, including *Ficolin 3* (*FCN3*), which is an essential component of the lectin complement pathway of the immune system; *Ribosomal Protein S6 Kinase A1* (*RPS6KA1*), which has been implicated in activating Toll-like receptor (TLR) 4 signaling; *High Mobility Group Nucleosomal Binding Domain 2* (*HMGN2*), which has known antimicrobial activity against bacteria, viruses, and fungi; and *Zinc Finger 683* (*ZNF683*), a tissue-resident T cell transcription regulator [35,36,37,38]. These promising candidate genes underlying fore udder attachment reflect activity related to both the immune response and the physical trait, reinforcing the role of this udder type trait as a major risk factor for mastitis.

### 4.3. Rear Teat End Shape

We previously associated the rear teat end shape with increased odds of both an elevated SCC and clinical mastitis in this cohort of cows [18]. In our current analyses, a similar pattern of candidate gene function emerged, reflecting activity related to both cell division (potentially explaining variation in morphology) and immune response regulation (reinforcing teat end shape as an appropriate indicator of mastitis risk). An associated SNP positioned at 26:50630351 resides within an intron of *Kinase Non-Catalytic C-Lobe Domain Containing 1* (*KNDC1*), which has been implicated in the regulation of cellular senescence and cell cycle progression [39]. In addition, *Adhesion G Protein-Coupled Receptor A1* (*ADGRA1*) was identified within a 1 Mb window of this SNP and belongs to a family of receptors known to regulate immune signaling [40]. We observed a pattern of candidate gene function like that of fore udder attachment, supporting the idea that selecting for cows on the basis of udder and teat conformational traits is an appropriate strategy for mastitis control.

### 4.4. Rear Teat Width

In this cohort of cows, we previously associated increasing rear teat width with greater odds of clinical mastitis [18]. Our case–control GWA comparing narrower rear teats (representing low mastitis risk) to wider rear teats (high mastitis risk) associated a single, novel QTL at BTA 25:38568564 in a gene-scarce region. In contrast, the linear GWA for the quantitative trait of rear teat width implicated candidate genes related to both cell division and immune function at many different QTL spanning the genome (Figure 5). We associated a novel SNP at BTA 11:104129366 and identified a nearby candidate gene *Caspase Recruitment Domain-Containing Protein 9* (*CARD9*), a key modulator of the immune response related to TLR and the NOD2 signaling pathway and shown to modulate microbiota to mediate pathogen susceptibility [41]. Furthermore, genes related to cell differentiation and survival were identified, including *Notch Receptor 1* (*NOTCH1*), a highly conserved protein with an extracellular domain containing many epidermal growth factor repeats and whose signaling is heavily involved in cell fate specification; *Epidermal Growth Factor-Like 7* (*EGFL7*), which is involved in Notch binding; and *Myomaker Myoblast Fusion Factor* (*MYMK*), which also resides in this region and has been associated with muscle hypertrophy, making it a strong candidate for impacting variation in teat morphology [42,43].

A QTL was associated at BTA 16:61802991-62196774 and has been previously mapped for milk casein and fatty acids [44,45]. Nearby candidate genes include *Centrosomal Protein 350* (*CEP350*), which plays a critical role in microtubule binding and spindle integrity during cell replication, and *Major Histocompatibility Complex Class I-Related* (*MR1*), which is important to the adaptive immune response [46,47]. A novel QTL was associated at BTA 16:63823597, and candidate gene investigation of this region revealed *Laminin Subunit Gamma 1 and 2* (*LAMC1/2*), which are thought to regulate cell organization into tissues, potentially contributing to variation in teat width [48]. This region is also home to genes related to immune function, including *Ribonuclease L* (*RNASEL*), which is involved in interferon regulation, and *DExH-Box Helicase 9* (*DHX9*), which has been found to control TLR-stimulated immune responses [49,50].

A significantly associated QTL at BTA 19:29058547-29063744 is located within a *Growth Arrest Specific 7* (*GAS7*) intron, which, while previously understood to influence neuron differentiation, was recently found to be abundantly expressed in murine alveolar macrophages, suggesting it may play a central role in mucosal immunity, though its exact roles in immune responses are still unknown [51]. This region may be highly pleiotropic, as in addition to its current implication in teat width, it was previously mapped for many economically important cattle traits, including feed intake, calving ease, milk components, ovulation rate, milk SCS, and blood immunoglobulin G levels [52,53,54,55,56,57]. Investigation of the associated novel QTL at BTA 19:22640468 revealed promising candidate genes, including *HIC ZBTB Transcriptional Repressor* (*HIC1*) and *Tyrosine 3-Monooxygenase Activation Protein* (*YWHAE*), which have both been associated with the regulation of cell proliferation, and the transcriptional regulation function of *HIC1* has been specifically tied to immune homeostasis at mucosal surfaces [58,59].

We associated a QTL at BTA 25:40126743-40190566, which resides within an intron of *Sidekick Cell Adhesion Molecule 1* (*SDK1*), an adhesion molecule isoform in the immunoglobulin superfamily primarily known for synapse formation in the retina, though Sidekicks in general are expressed in many different tissue types [60]. A previous study associated this QTL with residual feed intake, though it has not been implicated in mastitis or type traits until now [61]. A novel QTL at BTA 25:35208040 was associated with rear teat width, and interrogation of this region revealed a number of promising candidate genes, including *Cut Like Homeobox 1* (*CUX1*), which is known for its role in morphogenesis as well as regulating antigen presenting cells; *Myosin Light Chain 10* (*MYL10*), which is implicated in immune cell transmigration; and *Tripartite Motif Containing 56* (*TRIM56*), an ubiquitin ligase with a role in antiviral innate immunity [62,63,64,65]. Our direct phenotyping approach associated a novel QTL for rear teat width, implicating promising candidate genes that may provide biological insight into the regulation of this trait and reinforce its appropriateness as a proxy for mastitis.

### 4.5. Rear Udder Height

We previously associated low rear udder height with increased odds of clinical mastitis diagnosis in this cohort of cows, and a case–control GWA comparing low rear udders (representing high mastitis risk) to intermediate and high rear udders (representing low mastitis risk) associated a QTL at BTAs 6, 14, 15, 18, and 22 [18]. A significantly associated SNP at BTA 14:27024015 lay within an intron of *Aspartate Beta-Hydroxylase* (*ASPH*), known for hydroxylating epidermal growth factors and contributing to dysmorphic features; this region has not been associated with mastitis or type traits previously [66]. In addition, investigation of a significantly associated QTL at BTA 18:62273143-62481417 revealed nearby *Troponin T1 Slow Skeletal Type* (*TNNT1*), a sarcomere regulatory complex associated with muscle weakness, which could feasibly contribute to weak rear udder attachments and, consequently, low rear udder heights [67]. This QTL was previously associated with beef cattle calving ease but has not been implicated in mastitis or udder morphology until now [68]. Located in this same region was *NLR Family Pyrin Domain Containing 2* (*NLRP2*), a member of the NLR family known to regulate immune responses [69]. The combination of candidate genes related to both immunity and the regulation of morphology implicated in this GWA emphasizes the value of considering rear udder height in breeding and culling decisions.

### 4.6. Primiparous Subset

We hypothesized that the genotype by environment interaction may be greater in multiparous cows who have had greater mastitis exposure and mechanical manipulation of the udder and teats via milking. Time and the environment are both risk factors for mastitis, which could affect our analysis, given that udder conformation changes with age and any effects of farm and milking management will be more pronounced in older cows. To preclude these confounding factors, we evaluated a primiparous-only subset of cows (*n* = 144) for each trait. Only linear GWAs assessing continuous variation in front teat placement and udder depth significantly associated a QTL, likely due to insufficient power given the small samples size. The lack of success with the case–control approach is likely explained by a few risk phenotypes (<20%) and lower phenotypic variation, in general, observed among primiparous cows.

Regarding udder depth, different inheritance patterns and QTL were identified for the total cohort and primiparous subset, which is likely explained by farm culling impacting genotypic frequencies in these populations (Table 1; Figure 3). The total cohort GWA for udder depth associated a novel SNP at BTA 5:113268242 located within an intron of *Transcription Factor 20* (*TCF20*), which has been associated with human muscular dystrophy and postnatal overgrowth, such as tall stature, macrocephaly, and obesity [70]. No associated SNPs passed the Bonferroni correction in the primiparous subset GWA for udder depth, though a strong signal at BTA 17:34476230-36159480 included 11 SNPs passing the FDR, which were in linkage with *Fibroblast Growth Factor 2* (*FGF2*) and *Nudix Hydrolase 6* (*NUDT6*), both hypothesized to regulate cell proliferation [71]. In addition, in linkage with this QTL lay *Sprouty RTK Signaling Agonist 1* (*SPRY1*), which has been shown to influence mammary epithelial morphogenesis during post-natal development by negatively regulating epidermal growth factor signaling [72]. This QTL was previously mapped for teat length and milk yield but not for udder depth nor any mastitis indicator traits [73,74]. We posit that this QTL was masked in our analysis of the total cohort due to the differing genotypic and phenotypic frequencies in the total cohort, which was heavily exposed to the selective pressures of culling and the impact of the milking machine on the phenotype. Perhaps to elucidate the genetic mechanisms underlying these morphological characteristics, primiparous populations with minimal exposure to the environment must be evaluated.

## 5. Conclusions

This was the first study to use high-density SNP chip data with direct phenotyping and no reliance on imputation to investigate the genetic mechanisms regulating bovine udder and teat conformation and, by proxy, mastitis risk. For many traits, we significantly associated QTL spanning the genome, suggesting that udder and teat morphologies are complex traits resulting from many genes with small effect sizes. A noteworthy aspect of this study was the use of PCA to calculate a single, composite trait representing the overall mastitis risk on the basis of udder and teat type traits. This approach implicated novel QTL in mastitis susceptibility. Furthermore, hypothesizing that the genotype by environment interaction is large in multiparous cows, we evaluated a primiparous subset of cows whose minimal exposure to the environment (milking machine influence on phenotype, selection bias introduced by farm culling) may provide more refined biological insight into the genetics underlying udder and teat morphology. In particular, the objective of this study was to identify genomic loci that may contribute to the development of our previously identified risk factors of loose fore udder attachment, flat rear teat ends, and low rear udder height. Candidate genes surrounding QTL associated with these risk factors revealed functions related to both the immune response and cell proliferation and tissue morphology, reflecting their ability to represent mastitis susceptibility. This approach identified novel QTL for both mastitis and type traits, which may be valuable additions to national genetic evaluations for mastitis resistance.

## Figures and Tables

**Figure 1 animals-11-01147-f001:**
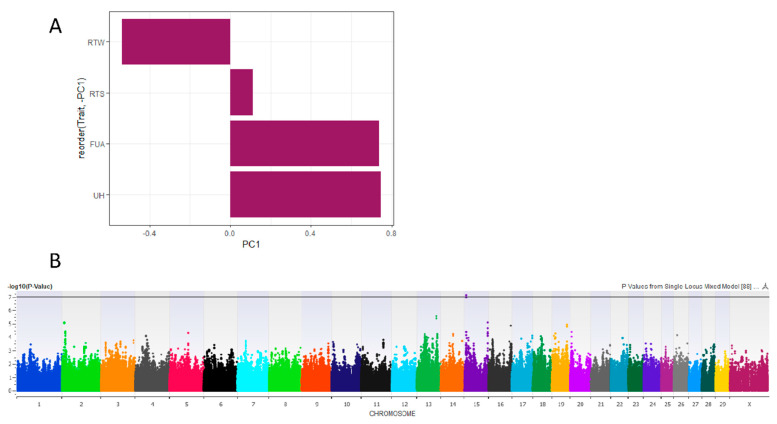
Creation of a new mastitis risk trait for genome-wide association (GWA) using principal component (PC) analysis. (**A**) Loading values for risk PC1 by rear teat width (RTW), rear teat end shape (RTS), fore udder attachment (FUA), and udder height (UH), where cows with higher risk PC1 scores likely have lower mastitis risk. (**B**) Manhattan plot showing -log_10_
*p*-values by chromosome in a linear GWA for risk PC1, with the black line representing the Bonferroni multiple testing correction threshold of 0.05.

**Figure 2 animals-11-01147-f002:**
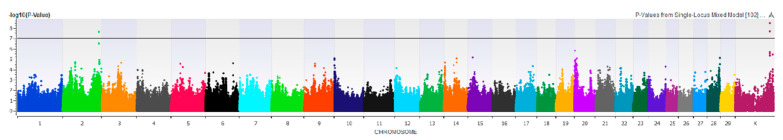
Genome-wide association (GWA) for fore udder attachment. Manhattan plot showing -log_10_
*p*-values by chromosome in a case–control GWA comparing loose fore udders to tight fore udders (*n* = 288). The black line represents the Bonferroni multiple testing correction threshold of 0.05.

**Figure 3 animals-11-01147-f003:**
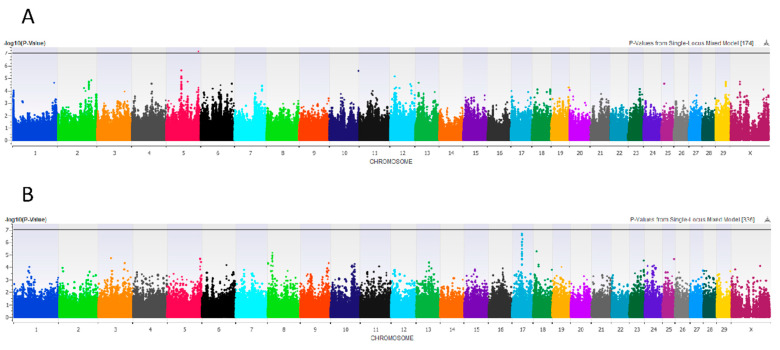
Genome-wide association (GWA) for udder depth. Manhattan plot showing -log_10_
*p*-values by chromosome in (**A**) a case–control GWA comparing deep (high mastitis risk) versus high udders among the total cohort (N = 458) and (**B**) a linear GWA with all udder depth scores (deep, intermediate, high) for a primiparous-only subset of cows (*n* = 144). The black line represents the Bonferroni multiple testing correction threshold of 0.05.

**Figure 4 animals-11-01147-f004:**
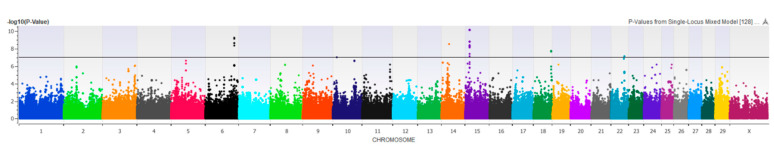
Genome-wide association (GWA) for rear udder height. Manhattan plot showing -log_10_
*p*-values by chromosome in a case–control GWA comparing low rear udders to all other udder height classifications (N = 458). The black line represents the Bonferroni multiple testing correction threshold of 0.05.

**Figure 5 animals-11-01147-f005:**
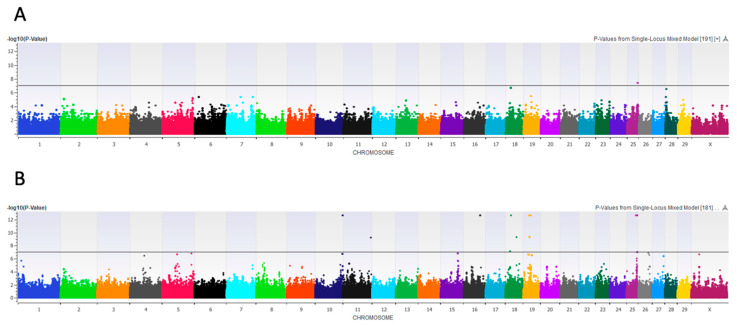
Genome-wide association (GWA)s for rear teat width. Manhattan plots showing -log_10_
*p*-values by chromosome in (**A**) a case–control GWA for the rear teat width, where cows are divided along the median value, representing extreme differences in morphology, and (**B**) a linear GWA for the raw measurements of the rear teat width, accounting for continuous variation in the trait. The black line represents the Bonferroni multiple testing correction threshold.

**Figure 6 animals-11-01147-f006:**
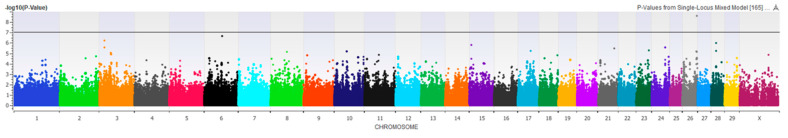
Genome-wide association (GWA) for rear teat end shape. Manhattan plots showing -log_10_
*p*-values by chromosome in a case–control GWA comparing flat (high mastitis risk) with pointed (low mastitis risk) rear teat ends. The black line represents the Bonferroni multiple testing correction threshold.

**Table 1 animals-11-01147-t001:** Models selected. Summary of final models, including sample size (N), model type, inheritance patterns, quality control measures, trait heritabilities, and total number of significantly associated SNPs.

Trait	N	Model Type	Inheritance	Pseudo-Lambda ^1^	Pseudo-Heritability	FDR ^2^	Bonferroni ^2^
Front teat length	458	Linear	Recessive	1.00	0.33	5	2
Front teat width	458	Case–control ^3^	Recessive	1.01	0.06	4	4
Fore udder attachment	288	Case–control ^4^	Additive	1.00	0.45	4	3
Risk PC ^5^ 1	458	Linear	Dominant	1.01	0.07	2	1
Rear teat length	458	Case–control ^6^	Additive	1.00	0.42	1	1
Rear teat end shape	227	Case–control ^7^	Recessive	1.02	0.50	1	1
Rear teat width	458	Case–control ^3^	Recessive	1.02	0.03	3	1
Rear teat width	458	Linear	Recessive	0.98	0.18	50	23
Udder depth	265	Case–control ^8^	Dominant	0.99	0.99	1	1
Udder height	458	Case–control ^9^	Recessive	1.00	0.02	129	32
Udder width	458	Case–control ^10^	Dominant	1.00	0.30	1	1
Front teat placement ^11^	144	Linear	Recessive	1.02	0.38	10	7
Udder depth ^11^	144	Linear	Recessive	1.05	0.62	10	0

^1^ Genomic inflation factor for model quality control, quantile–quantile plots in Appendix A; ^2^ number of SNP associations passing either Bonferroni or false discovery rate (FDR) multiple testing corrections at *p* < 0.05; ^3^ narrow versus wide teats, split by median measurement; ^4^ loose versus tight fore udders, excluding all cows with intermediate fore udder attachment; ^6^ short versus long rear teats, split by median measurement; ^5^ principal component; ^7^ flat versus pointed rear teat ends, excluding all cows with round rear teat end shape; ^8^ deep versus high udder depth, excluding all cows with intermediate udder depth; ^9^ low versus all other udder heights (intermediate and high rear udders combined); ^10^ wide versus all other udder widths (intermediate and narrow udders combined); ^11^ primiparous cow subset.

## Data Availability

The phenotypic data presented in this study are openly available at https://github.com/AshaMilesPhD/U-T_GWAS. Genomic data are available from corresponding author upon reasonable request.

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
