# Peer review of "Direct Phenotyping and Principal Component Analysis of Type Traits Implicate Novel QTL in Bovine Mastitis through Genome-Wide Association"

_animals, 2021, doi:10.3390/ani11041147_

Round 1

Reviewer 1 Report

The manuscript “Principal component analysis of type trait risk factors in Holstein cows helps identify plausible pleiotropic genomic loci contributing to mastitis” identified some novel candidate SNPs/genes for mastitis disease. The population size is small (N =471) comparing to the current stage of the GWAS in mastitis. However, the manuscript proposed some novel approaches and benefited from the high-density SNP chip; therefore, the present results are essential for the field. The manuscript is well written, and the results are well discussed. I suggest the authors extend the introduction to the state of the art of genomic studies in mastitis and related disease.

Minor comments:

Line 36; Genome-wide associations were performed: Might add the method for GWAS

Line 48, 60: The authors might consider the consistent ways for writing the USA, US, or United States

Line 85: GWA should be used in line 66 as the first appear in the manuscript.

Line 70-71:  The authors might add a summary about the QTL and genes that have been  identified for mastitis,

Line 96: Change 4 to four

Line 185: why did the authors choose a very stringent threshold for multiple testing correction in the current population instead of using FDR?

The Manhattan plot showed many SNPs for the X chromosome. Why did the author include the X chromosome in the analyses? And how many markers in the X chromosome?

There are two many Manhattan plots in the manuscript, and I suggest the authors add the summary of the most critical SNPs/genes in a table, which could help the reader refer to the findings easier. The authors might also combine all these plots in one or two figures, making it easier to check.

Line 371: Should write as TLR4 (Toll-like Receptor 4) to keep the consistent

Author Response

The authors thank the reviewers for their efforts and helpful comments. Responses to their specific comments are detailed below. Changes to the manuscript are highlighted in yellow in the attached file. Please note that after the editorial office’s formatting changes the in-text equations are not in the correct positions. We have moved those equations so they are located where they are referenced and explained in the methods.

Reviewer 1

The manuscript “Principal component analysis of type trait risk factors in Holstein cows helps identify plausible pleiotropic genomic loci contributing to mastitis” identified some novel candidate SNPs/genes for mastitis disease. The population size is small (N =471) comparing to the current stage of the GWAS in mastitis. However, the manuscript proposed some novel approaches and benefited from the high-density SNP chip; therefore, the present results are essential for the field. The manuscript is well written, and the results are well discussed. I suggest the authors extend the introduction to the state of the art of genomic studies in mastitis and related disease.

AU: Thank you for these suggestions. Comprehensive reviews on mastitis and their related diseases, dairy genetic selection & genomic tools, and mastitis genetics are abundant in the literature and we prefer to point the readers towards those papers rather than repeat their words here (e.g., Ruegg 2017 DOI: 10.3168/jds.2017-13023, Weigel et al., 2017 DOI: 10.3168/jds.2017-12954, Miles and Huson 2020 DOI: 10.3168/jds.2020-18297). We have added a sentence referring to these papers L50-52.

Minor comments:

Line 36; Genome-wide associations were performed: Might add the method for GWAS

AU: Per the author guidelines (https://www.mdpi.com/journal/animals/instructions#manuscript), the word limit for the abstract is 200 words. Our current abstract is 224 words, so we prefer to let the reader find details regarding the GWAS in our methods section in order to adhere to the word limits as closely as possible.

Line 48, 60: The authors might consider the consistent ways for writing the USA, US, or United States

AU: Thank you for this suggestion. As with other abbreviations, we prefer to spell out United States at first mention (L48). All subsequent mentions use the same abbreviated format of U.S. (e.g., L62, L76, and L87).

Line 85: GWA should be used in line 66 as the first appear in the manuscript.

AU: We have moved our definition of GWA from L85 to L68 as suggested.

Line 70-71:  The authors might add a summary about the QTL and genes that have been identified for mastitis,

AU: This is an important point and addressed in detail in our discussion when we place our results in the larger context of the field. We did not repeat this information in the introduction to avoid fatiguing the reader.

Line 96: Change 4 to four

AU: This has been revised as suggested L97.

Line 185: why did the authors choose a very stringent threshold for multiple testing correction in the current population instead of using FDR?

AU: As the reviewer noted, this is a relatively small sample size for cattle GWA studies. While we believe these detailed phenotypes (which logistically can only be collected on a smaller scale) provide valuable insight into the genetic regulation of these traits, we understand this can be a limitation. We chose a strict multiple testing correction threshold in acknowledgement of our small sample size and the multiple methods used to assess each trait, as is explained in L333-337.

The Manhattan plot showed many SNPs for the X chromosome. Why did the author include the X chromosome in the analyses? And how many markers in the X chromosome?

AU: While the X chromosome is still not well annotated/understood, it has become standard to include it in genetic studies. The CattleQTLdb reports at least 25,818 QTL/associations located on the bovine X Chromosome (https://www.animalgenome.org/cgi-bin/QTLdb/BT/browse); we included it in our analyses to be consistent with the standards of the field. Regarding your question of the number of markers on BTX, the new assembly (ARS UCD 1.2) reports the X chromosome 6.6% longer than UMD3.1 for a total length of 139,009,144bp (https://interbull.org/static/web/11_15_Null.pdf, https://www.ncbi.nlm.nih.gov/assembly/GCA_002263795.2/#/st). The ARSUCD1.2 assembly includes 34,145 SNPs on the X Chromosome.

There are two many Manhattan plots in the manuscript, and I suggest the authors add the summary of the most critical SNPs/genes in a table, which could help the reader refer to the findings easier. The authors might also combine all these plots in one or two figures, making it easier to check.

AU: Animals does not have any limit on manuscript length or figures, but we agree too many Manhattan plots can be cumbersome to the reader. To limit the number of figures in the paper (6 total), we only included Manhattan plots for the risk traits emphasized. The remainder are included in Supplementary Materials for readers to review as preferred while keeping the manuscript uncluttered. We prefer Manhattan plots to gene summaries because they visually convey the strength of the signal at each QTL. Per reviewer 2’s request, we did add additional information to Supplementary Table 1, making it easier for the reader to quickly identify novel loci and whether they were implicated in morphology or immunology.

Line 371: Should write as TLR4 (Toll-like Receptor 4) to keep the consistent

AU: Recognizing Animals has a diverse audience, Toll-like Receptor is defined at first use at L372, and TLR already used consistently everywhere else.

Reviewer 2 Report

The title does not reflect the content. A suggestion, an alternative: Combinations of udder and teat traits and contrasts in their scores in Holstein cows and heifers identify genomic loci involved in mastitis susceptibility.

In this paper new and old loci involved in morphology and immunology are detected in an original approach. Clear proves for pleiotropy are not found nor discussed in the paper.

Line 21-23: When you mentions the opportunity for selection you should also discuss and conclude how this can be implemented.

Line 54: typing error heath instead of health

Line 88: what are robust phenotypes? What does robust means?

Line 159: What are the fixed effects? Are the two commercial farms treated as fixed?

Line 164-166: the implications of these inheritance models should be discussed after Table 1 and/or in the discussion.

Somewhere, at least before the results an illustration of an udder of a cow or cows that illustrates the morphology of the different phenotypes might help a reader who is not familiar with type scoring of cows.

Line 201: mention the specific PCs here.

Table 1: see comment line 164-166

Line 312-314: I do not see this in Figure 6

The results found in this study can be much clearly presented in a table with e.g. at least the two columns: loci involved in morphology and loci involved in immunology (and if necessary a third one: traits involved in other processes. The new ones can be presented in bold.

Line 469: you should describe the environment and the interaction here: the effects of housing system (bedding), feeding system, milking machine and milking practices etc. will be more pronounced in older cows because they are longer exposed to these practices. The udder of a cow changes as cows grow older: udder depth and front teat changes show most pronounced changes affecting the proper operation of a milking machine. A risk factor for mastitis.

In conclusion: an original and scientifically sound paper. The authors can help the reader (especially the one less familiar with dairy cattle) with an illustration of udder traits and with a table presenting the loci identified.

Author Response

The authors thank the reviewers for their efforts and helpful comments. Responses to their specific comments are detailed below. Changes to the manuscript are highlighted in yellow in the attached file. Please note that after the editorial office’s formatting changes the in-text equations are not in the correct positions. We have moved those equations so they are located where they are referenced and explained in the methods.

Reviewer 2

The title does not reflect the content. A suggestion, an alternative: Combinations of udder and teat traits and contrasts in their scores in Holstein cows and heifers identify genomic loci involved in mastitis susceptibility. In this paper new and old loci involved in morphology and immunology are detected in an original approach. Clear proves for pleiotropy are not found nor discussed in the paper.

AU: Thank you for these comments. We agree that investigating pleiotropy is not a major emphasis of this work so have updated the title accordingly.

Line 21-23: When you mentions the opportunity for selection you should also discuss and conclude how this can be implemented.

AU: We think elaborating on genetic selection is outside of the scope of the Simple Summary but agree more details could be beneficial. This was explained in L95-98, and L509-516.

Line 54: typing error heath instead of health

AU: Thank you for catching this, revised as suggested L56

Line 88: what are robust phenotypes? What does robust means?

AU: By “robust” we mean strong, reliable characterization. We have changed this adjective to “detailed” (L90) to avoid confusion with “phenotypic robustness” which is commonly used when talking about evolution and not what we are referring to.

Line 159: What are the fixed effects? Are the two commercial farms treated as fixed?

AU: Farm is among the fixed effects considered in each model. This is explained L166 and we have changed “variables” to “fixed effects” to clarify this point.

Line 164-166: the implications of these inheritance models should be discussed after Table 1 and/or in the discussion.

AU: We have added an explanation of additive (risk of phenotype increases with each copy of allele), dominant (only 1 copy of allele is required for increased risk), and recessive (2 copies must be present for increased risk of phenotype) genetic models to the methods. These are common genetic concepts and to avoid over-capitulating, we have added a citation L166 which directs readers to a published paper from a genetics course which covers the basics of inheritance models if they desire more information on this topic.

Somewhere, at least before the results an illustration of an udder of a cow or cows that illustrates the morphology of the different phenotypes might help a reader who is not familiar with type scoring of cows.

AU: Thank you for suggesting we make this more accessible to a diverse audience. We have included a direct link to the Holstein Associations guidelines L115-116, where they have many pictures explaining the scoring system. Because this is copyrighted material we cannot include the images in the paper itself.

Line 201: mention the specific PCs here.

AU: Our PCA generated a total of 30 PCs which were evaluated. The criteria used to evaluate and select PCs were explained L131-134, and the annotated code and data used to generate them is publicly available so the reader may recreate them if they wish. We did not discuss the other 28 PCs which are not informative nor analyzed to avoid confusing the reader and adding unnecessary bulk to the paper.

Table 1: see comment line 164-166

AU: See response re: L164-166

Line 312-314: I do not see this in Figure 6

AU: See L314, this refers to Supplementary Figure 6, not Figure 6.

The results found in this study can be much clearly presented in a table with e.g. at least the two columns: loci involved in morphology and loci involved in immunology (and if necessary a third one: traits involved in other processes. The new ones can be presented in bold.

AU: Associated loci and their candidate genes are already reported in Supplementary Table 1. As detailed in our discussion, most loci are implicated in both morphology and immunology so we aren’t sure that a new table would clarify anything. The objective of this study (L95-98) was not to discover and describe gene function/biological pathways, but to identify plausible SNP markers for inclusion in genomic selection marker panels for mastitis-resistant cows. Candidate gene function is discussed to assess the relevance of these markers and as an additional safeguard against spurious associations which likely do not have bearing on mastitis. However, we agree the reader might benefit from a quick-reference of loci and their implicated pathways, so we have added an additional column to Supplementary Table 1 indicating each loci’s association with morphology, immunology, or other. Bold face is already being used to indicate genes containing the associated SNP; novel traits are indicated with an asterisk after their coordinates.

Line 469: you should describe the environment and the interaction here: the effects of housing system (bedding), feeding system, milking machine and milking practices etc. will be more pronounced in older cows because they are longer exposed to these practices. The udder of a cow changes as cows grow older: udder depth and front teat changes show most pronounced changes affecting the proper operation of a milking machine. A risk factor for mastitis.

AU: We elaborated on this point L470-476.

In conclusion: an original and scientifically sound paper. The authors can help the reader (especially the one less familiar with dairy cattle) with an illustration of udder traits and with a table presenting the loci identified.